# *Quality > Quantity*: Synthetic Corpora from Foundation Models for Closed-Domain Extractive Question Answering

## Abstract

Domain adaptation, the process of training a model in one domain and applying it to another, has been extensively explored in machine learning. While training a domain-specific foundation model (FM) from scratch is an option, recent methods have focused on *adapting* pre-trained FMs for domain-specific tasks. However, our experiments reveal that either approach does not consistently achieve state-of-the-art (SOTA) results in the target domain. In this work, we study extractive question answering within closed domains and introduce the concept of *targeted pre-training*. This involves determining and generating relevant data to further pre-train our models, as opposed to the conventional philosophy of utilizing domain-specific FMs trained on a wide range of data. Our proposed framework uses Galactica to *generate* synthetic, "targeted" corpora that align with specific writing styles and topics, such as *research papers* and *radiology reports*. This process can be viewed as a form of *knowledge distillation*. We apply our method to two biomedical extractive question answering datasets, COVID-QA and RadQA, achieving a new benchmark on the former and demonstrating overall improvements on the latter. Code available upon publication.

## 1 Introduction

Our work revolves around three key pillars: Extractive Question Answering (EQA), Domain Adaptation, and knowledge distillation through *prompting* generative foundation models (FMs). EQA, a long-standing problem in natural language processing (NLP) involves identifying a token span in a text passage to answer a given question. The task is typically evaluated using datasets like SQuAD (Rajpurkar et al., 2016) and DuoRC (Saha et al., 2018). While recent architectures like BERT (Devlin et al., 2019), T5 (Raffel et al., 2020), and GPT-3 (Brown et al., 2020) have made remarkable advancements in this task, their performance suffers when applied to domain-specific data, especially in the biomedical/clinical domain (Moradi et al., 2021).

The performance discrepancy in models is linked to the definition of a *domain*, i.e., the loose NLP equivalency of $domain = genre$ or *thematic content* of a dataset. This definition is quite restrictive (Plank, 2016). Ideally, a model pre-trained on a specific theme should excel in tasks related to that subject matter. However, not all domain-specific models are equal as illustrated by the differing performances of BioBERT (Lee et al., 2020) and PubMedBERT (Gu et al., 2021), even though both trained on PubMed data. We suggest redefining $domain = [genre + dataset]$, emphasizing the importance of tailoring the training data to the subject matter of the task. This approach acknowledges that a *one-domain-model-to-rule-them-all* is not universally applicable, and the learning should focus on concepts relevant to specific tasks. We define "closed-domains" as *datasets* related to highly specialized subjects like medicine, law, or finance.

The third pillar supporting our work is the recent progress in generative FMs (Ye et al., 2023; OpenAI, 2023). While ChatGPT performs well on the USMLE (Kung et al., 2023), our experiments demonstrate that large, general-domain (and even closed) FMs struggle with tasks involving highly specialized language, such as COVID-QA (Möller et al., 2020) and RadQA (Soni et al., 2022). Additionally, their autoregressive architecture is not well-suited for extractive QA as they are designed to *synthesize* new text rather than *extract spans* from given text (c.f. sec. 3.1). Also, when presented with sequences exceeding the model's context length, they need to be divided into overlapping segments. Although this challenge applies to both bi-directional and generative models, bi-directional models are more suitable due to their inherent capabilities. While a generative model can generate an answer for each segment, it lacks the ability to indicate the model's confidence in each answer, a

feature provided by bi-directional models.

To overcome these limitations, we propose distilling the knowledge from generative FMs into smaller, bi-directional language models (LMs) better suited for EQA. We leverage recent breakthroughs in FMs and architectures better suited for the task. Our approach involves using a generative FM to generate a synthetic corpus tailored to a specific application and fine-tuning a bi-directional, general-purpose LM on this corpus. The results of our approach demonstrate the efficacy and running time improvements as compared to existing domain-specific LMs.

In the seminal work in this area, West et al. (2022) demonstrate how GPT3 could be utilized to create high-quality knowledge graphs via prompting. He et al. (2022) show how a GPT model could be used as a "teacher" to distil knowledge into a "student." Similarly, Peris et al. (2022) used unlabelled task-relevant data and trained multilingual students with varying proportions of general/task-specific data and report the most gains using "only the downstream task's unlabelled data".

Gururangan et al. (2020) introduces the concepts of DAPT (Domain-Adaptive Pretraining) and TAPT (Task-Adaptive Pretraining), which are similar to our approach. DAPT involves extended pre-training on domain-specific corpora without labels, while TAPT focuses on pretraining on the unlabelled training set of the downstream task. Although they demonstrate the effectiveness of TAPT compared to DAPT, closed-domain datasets like COVID-QA typically lack a separate unlabelled training set and may not even have train/dev/test splits. Further, DAPT considers knowledge beyond what is specifically relevant to the task data, whereas our approach confines training to the required concepts.

In summary, **our contributions** are (a) proposing a pipeline for generating customized pre-training data for closed domains, (b) demonstrating the effectiveness of synthetic data in achieving substantial gains with reduced memory footprint, (c) showcasing the benefits of creative prompting and dataset awareness, (d) setting a new benchmark on COVID-QA & overall improvements on RadQA.

## 2 Methodology

In Figure 1, we present our method and compare it to existing pre-training paradigms. The current approaches involve training a randomly initialized architecture from scratch (top) on either open-domain data (e.g., BERT/RoBERTa (Liu et al., 2019)) or closed-domain data (e.g., SciBERT (Beltagy et al., 2019), PubMedBERT (Gu et al., 2021)), or adopting an extended pre-training approach (middle), where the model is initially trained on open-domain data and then further pre-trained on unlabelled domain-specific text (e.g., BioBERT) to adapt it to the closed-domain. The former emphasizes stronger domain representations, while the latter prioritizes computational efficiency by not requiring the model to learn a general sense of language. After training, these models typically require fine-tuning on datasets like SQuAD to learn the task, and can undergo additional fine-tuning for domain adaptation on the final dataset.

While the above techniques have achieved much success, they typically rely on high quantities of unlabelled corpora to yield useful results, thus raising the question: *What happens when we do not have enough "relevant" domain data, either in style or volume*? To this end, we introduce the notion of **targeted pre-training**, which focuses on a specific subset of the domain, tailor-made for the ultimate downstream dataset.

Our method works as follows. First, we **combine all the questions and contexts** from the training split of the EQA dataset. Unfortunately, COVID-QA does not have a train-dev-test split. In such a situation, we consider the entire dataset for the next step (we test for cheating/information leakage in this case as described in sec. 4.3). Next, we **extract entities** through Named Entity Recognition (NER) using scispaCy (Neumann et al., 2019). Comparing the *small* and *large* versions of the NER models, we found the former (en_core_sci_sm) yields qualitatively better & quantitatively more, entities.

Next, we **create prompts** for the identified entities to generate contexts mimicking the respective datasets. This required studying the characteristics of the datasets such as the style of contexts (full research articles in COVID-QA & radiology reports in RadQA), their lengths and relevant keywords. The collection of prompts were then supplied to Galactica (Taylor et al., 2022), to **generate the corpora** ($\cup$ generated contexts) for pre-training.

Galactica is a decoder-based FM pre-trained on a collection of text encompassing research articles, knowledge bases, code and even LaTeX markup. Galactica is equipped with the feature of being able to generate research papers by being prompted as

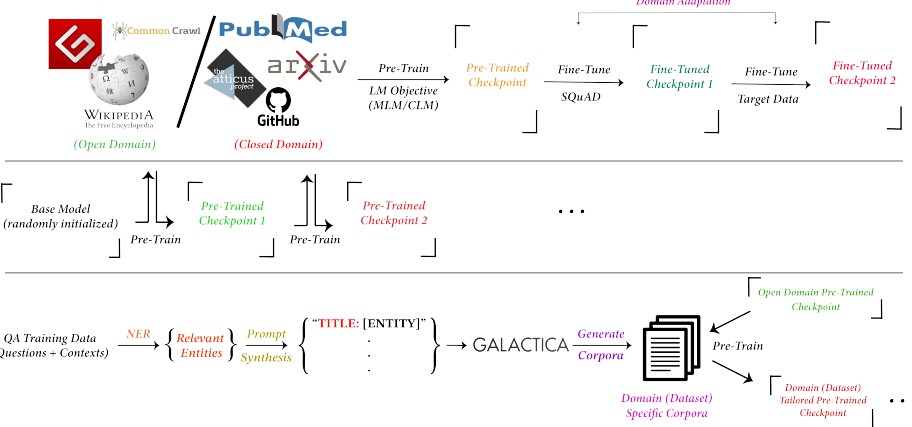

Figure 1: Pre-Training Pathways: From scratch (top); Extended (middle); **Targeted** (bottom; ours) | Note: We only show Fine-Tuning on EQA as it is the task of interest | The prompt handle is written in CAPITAL for emphasis.

"`Title: [entity]`" (where `Title:` is the prompt handle/keyword and `entity` is the entity for which we require generated content). We considered other generative models such as BLOOM (Scao et al., 2022) and PubMedGPT. However, they were either producing multilingual text for our prompts (former) or their generations were qualitatively inferior to Galactica (both).

The choice of prompt for COVID-QA is straightforward (as above) seeing as its contexts are research articles themselves. RadQA, on the other hand, presents a bigger challenge. Its contexts are redacted radiology reports without any consistent format (Hartung et al., 2020). This proved to be a challenge since we did not have a template for which to synthesize prompts. However, after going through the samples in the dataset, we realized that the Findings and Impressions section are the most vital in a patient's report (akin to the experiment and results section in a research paper). Such clues led us to construct our prompt for RadQA as, "`Patient has [entity]. FINDINGS AND IMPRESSION`". This was very interesting for us since Galactica had never seen radiology reports during training and we found a way to get it to synthesize *pseudo-reports* in this manner bypassing any privacy concern. We specifically wrote our prompt in this way so as to acquire text for both sections in a single go (for computational efficiency) and, to avoid *chain-of-thought-reasoning* (CoT) since we were using the base variant of Galactica (1.3B) which according to Wei et al. (2022), would not be able to keep track of logic seeing as it's $<<$ ~100B parameters.

After generating contexts, we perform extended

pre-training i.e., taking an open-domain pre-trained checkpoint (BERT/RoBERTa) and further training it on our generated corpus followed by two rounds of fine-tuning (SQuAD → COVID-QA/RadQA). A natural question to ask is why we *generated* a corpus rather than using existing text. We do this for 3 reasons, (a) *flexibility* to create content of a certain style, as mentioned before (b) some corpora can be *unavailable* due to privacy reasons or blocked behind paywalls, such as the corpora used by Gururangan et al. (2020), & (c) our tests can be used to determine if the content produced by such FMs is factually grounded and is able to teach the student models specific writing styles.

## 3 Experiments

Our study focuses on two datasets: COVID-QA, comprising 2,019 answerable QA pairs (no train/dev/test splits) sourced from CORD-19 (Wang et al., 2020), and RadQA, consisting of 6,148 QA pairs from radiology reports, with a train/dev/test split of 4,878/656/614. We conduct experiments in two primary areas: benchmarking and targeted pre-training.

### 3.1 Benchmarking

We identify ten encoder models to apply to each dataset. The application to COVID-QA required a domain-related model checkpoint fine-tuned on SQuAD v1 while RadQA contains questions with no answers and requires models fine-tuned on SQuAD v2 (Rajpurkar et al., 2018). For consistency, we utilized the *cased*, *base* version of each architecture when available. Models applied to COVID-QA were fine-tuned using five-fold cross-

Table 1: Benchmarking Bio Models (RadQA). H(F1): HasAns_F1, H(EM): HasAns_EM; *: ("18% papers from the computer science domain and 82% from the broad biomedical domain" (Beltagy et al., 2019)); Unified Medical Language System (UMLS); #: from U.S. Department of Veterans Affairs health care systems; †: Trained on UMLS KG for entity representations; MIMIC: Medical Information Mart for Intensive Care; S2ORC: The Semantic Scholar Open Research Corpus; Blue/red indicates best/worst scores; [1](Yan et al., 2022); [2](Alsentzer et al., 2019); [3](Gururangan et al., 2020)

| Model | Corpus | Corpus Size | Dev | | | | Test | | | |
|---|---|---|---|---|---|---|---|---|---|---|
| | | | EM | F1 | H(EM) | H(F1) | EM | F1 | H(EM) | H(F1) |
| **BioBERT** | PubMed | 4.5B words | 26.98 | 44.33 | 41.65 | 68.42 | 50.49 | 63.53 | 46.74 | 64.15 |
| **SciBERT** | Semantic Scholar* | 3.2B words | 26.68 | 44.34 | 40.94 | 68.21 | 53.26 | 67.91 | 47.83 | 67.38 |
| **PubMedBERT** | PubMed | 3.1B words / 21GB | 31.55 | 48.15 | 48.24 | 73.86 | 54.4 | 68.5 | 49.35 | 68.17 |
| **BlueBERT** | PubMed + MIMIC | 4.5B words | 31.55 | 48.02 | 48.24 | 73.65 | 54.23 | 67.83 | 48.91 | 67.07 |
| **CODER** | UMLS | N/A† | 40.24 | 57.47 | 45.41 | 72 | 52.93 | 67.96 | 50.43 | 70.49 |
| **LUKE** | Wikipedia | 3.5 billion words | 27.29 | 44.39 | 42.12 | 68.51 | 49.51 | 62.92 | 46.3 | 64.2 |
| **RadBERT**[1] | Radiology reports# | 2.6 GB | 32.01 | 49.57 | 46.11 | 73.22 | 51.14 | 65.36 | 50.65 | 69.64 |
| **ClinicalBERT**[2] | MIMIC | 0.5B words / 3.7GB | 28.35 | 44.51 | 43.76 | 68.7 | 50.17 | 63.39 | 46.74 | 64.38 |
| **BioMed-RoBERTa**[3] | S2ORC | 7.55B tokens / 47GB | 28.66 | 46.17 | 44.24 | 71.27 | 52.28 | 66.45 | 48.91 | 67.82 |
| **Galactica** | c.f. section 2 | 106B tokens | 1.37 | 8.5 | 1.37 | 8.5 | 0.49 | 10.23 | 0.49 | 10.23 |
| **MedLLaMA** | Medical Corpora | NA | 0.3 | 10.63 | 0.3 | 10.63 | 0.16 | 12.14 | 0.16 | 12.14 |
| **MedAlpaca** | Medical Meadow | NA | **1.68** | **15.18** | **1.68** | **15.18** | **1.3** | **16.95** | **1.3** | **16.95** |

validation and the resulting average performance across folds is presented in Table 3. Results of models applied to the prescribed splits are presented in Table 1. The metrics used are exact match (EM), binary measure of whether the prediction & gold spans are identical & F1, the harmonic mean of the number of shared words in the two spans w.r.t number of words in the prediction (precision) and w.r.t number of words in the gold span (recall).

To assess the **zero-shot** performance of three decoder models, namely Galactica-base (1.3B), MedLLaMA, and MedAlpaca (both 13B), we measure their ability to generate answers without further fine-tuning on our datasets, considering that decoders do not extract spans, but generate answers for comparison to gold spans. We selected Galactica-1.3B for consistency with our corpus generation experiments, MedLLaMA as a strong open-source medical checkpoint, and MedAlpaca as a medical QA-specific LLaMA checkpoint. Each sample was formatted as `Question:<question_text> Context:<part_of_context> Answer:` and the text generated after `Answer:` was considered as the predicted span. Due to the large size of COVID-QA contexts, they were segmented as they exceeded the maximum sequence length of each model (2,048 tokens). We report overall EM/F1 on each dataset and average best EM/F1 (parenthesis in Table 3) from each Q+C+A chunk for COVID-QA (N/A for RadQA since the context size was $<<$ models maximum input length).

## 3.2 Targeted Pre-training

Targeted pre-training begins by identifying named entities in each of our datasets. scispaCy `en_core_sci_sm` identifies roughly 47k and 11k named entities in COVID-QA and RadQA, respectively. Next, Galactica is used to generate contexts for the identified entities, constituting the synthetic dataset used for targeted pre-training. To maintain size-parity, five contexts are generated for each entity identified in RadQA, yielding around 55k total contexts. Galactica is allowed to use its full context size of 2,048 tokens to generate the synthetic data for each entity.

### 3.2.1 Corpus Size

When training models for COVID-QA, we investigated the impact of synthetic dataset size on downstream performance. We examined the effects of generating one context per entity and also explored generating ten contexts per entity, resulting in a dataset that was 10 times larger than the baseline. This analysis allowed us to assess the scalability of our proposed approach.

### 3.2.2 Context Length

The average context length for COVID-QA is 6k tokens, and Galactica has a maximum context size of 2k, resulting in a misalignment between the synthetic corpus and the target dataset. Increasing the context size of Galactica would mean training

it from scratch with architectural changes which is infeasible. Thus, we explore the impact of sequence length in the synthetic corpus by limiting the records to only 1k tokens. While we cannot determine if *longer* sequences are *beneficial*, we can evaluate if *shorter* ones are *detrimental*.

### 3.2.3 Token Filtering

We performed entity filtering as a common ablation technique for both datasets. We used regular expressions to remove entities with special characters such as \*, !, etc., as well as specific text patterns like `https*` and `baby`. We implemented a length-based filter, retaining only entities longer than a certain number of characters. Additionally, for COVID-QA, we applied a second round of filtration using TF-IDF, considering the questions + context as the corpus and retaining the top 25k entities with the highest IDF scores. However, as this approach did not yield satisfactory results, we decided not to use it for RadQA. Due to the large number of possible combinations, we did not extensively explore these settings in our experiments.

### 3.2.4 Prompting Style

We explore the use of two different prompts when encouraging Galactica to generate *pseudo* radiology reports - "`Patient has [entity]. FINDINGS AND IMPRESSION`" as described above, and simply "`[entity].`" Galactica was **not** pretrained on radiology reports, so the ideal prompt is not immediately obvious. In trying different options, we hope to find a satisfactory prompt.

### 3.2.5 Human-Generated Contexts

We established a *Wikipedia* baseline alongside our domain-specific pre-trained models to assess the influence of content and text structure during domain adaptation. Instead of utilizing Galactica to generate our corpus, we queried Wikipedia and retrieved the complete page associated with the top search result for each entity. This analysis aimed to gauge the significance of text content and structure. Wikipedia was chosen as it has been extensively used in the training data of varied models offering reliable information. The number of entities available for this baseline was $<<$ than to our approach since most of them do not exist in Wikipedia due to either being extremely esoteric, e.g., `pulmonary parenchymal infiltrate` or improperly formed, e.g., `Bao &`.

## 4 Discussion

Here we discuss the results of benchmarking existing models (Tables 1 & 3) as well as results for our targeted pre-training (Tables 2 & 4).

### 4.1 Baseline Analysis

#### 4.1.1 COVID-QA

Our benchmarking trials demonstrate that a one-size-fits-all approach does not work for domain adaptation. BioBERT and PubMedBERT were trained on similar corpora and yet yield similar performance, indicating no clear winner.

Surprisingly, the SciBERT (+CORD-19) checkpoint, trained on CORD-19 articles, performs worse than regular SciBERT, suggesting potential issues in training choices or noisiness in the data. **Notably, LUKE, trained solely on Wikipedia data, emerges as the best baseline model**, possibly due to its entity-recognition pre-training objective, which aids in identifying relevant entities for QA tasks (Van Aken et al., 2019). XLNet, degrades completely on COVID-QA, potentially due to the permutation of input tokens hindering its reasoning across large contexts.

#### 4.1.2 RadQA

**RadQA benchmarks were a bit less unanimous**. On the dev set, **CODER had the best overall EM & F1 but suffered a bit w.r.t PubMedBERT on only answerable questions**. This was not surprising since CODER is an extended PubMedBERT checkpoint trained to learn clinical embeddings from the UMLS knowledge graph which covers several terms found in radiology reports. Learning them led to an overall improvement of 27.54% EM & 19.36% F1 respectively.

PubMedBERT and BlueBERT exhibit similar performance on both development and test sets, which is unexpected considering that BlueBERT was pretrained on clinical notes from the MIMIC corpus. Surprisingly, RadBERT, despite being a RoBERTa architecture, outperforms PubMed/BlueBERT. Although RadBERT's performance shows slight enhancements, it was trained on smaller amount of data compared to others. This highlights the significance of domain alignment in terms of the data on which models are trained.

Unfortunately, LUKE performed poorly compared to Bio/Sci-BERT, showing little (dev) to no gain (test) in the evaluation. The impact of writing styles in the training corpora is evident in the

Table 2: Targeted Pre-training Results (RadQA). H(F1): HasAns_F1, H(EM): HasAns_EM; *: [Vanilla Fine-Tuning]. †: normal prompting, ‡: fancy prompting, ♣: entity filter. Blue/red indicates best/worst scores.

| Model | Time | Corpus Size / Train dataset | Dev | | | | Test | | | |
|---|---|---|---|---|---|---|---|---|---|---|
| | | | EM | F1 | H(EM) | H(F1) | EM | F1 | H(EM) | H(F1) |
| **BERT** | NA* | NA | 24.85 | 43.34 | 38.35 | 66.89 | 45.77 | 58.63 | 41.74 | 58.91 |
| **RoBERTa** | | | 26.37 | 44.26 | 40.71 | 68.31 | 50.81 | 64.38 | 47.61 | 65.71 |
| **BERT** | ≈30 mins | 18.4 MB/Wikipedia | 24.7 | 43.15 | 38.12 | 66.6 | 46.91 | 59.91 | 41.74 | 59.1 |
| **RoBERTa** | | | 26.98 | 45.28 | 41.65 | 69.88 | 50.17 | 63.5 | 48.04 | 65.85 |
| **BERT** | ≈11 hrs | 81.6 MB/ Galactica(≈55k) † | 24.7 | 42.51 | 38.12 | 65.61 | 47.39 | 59.56 | 42.61 | 58.84 |
| **RoBERTa** | | | 27.59 | 44.72 | 42.59 | 69.02 | 51.95 | 65.28 | 47.39 | 65.18 |
| **BERT** | ≈11 hrs | 80.3 MB Galactica(≈55k) †♣ | 25 | 43.24 | 38.59 | 66.74 | 47.88 | 60.06 | 42.83 | 59.08 |
| **RoBERTa** | | | 26.83 | 44.57 | 41.18 | 68.57 | 51.47 | 65.04 | 49.35 | 67.47 |
| **BERT** | ≈11 hrs | 38.1 MB/ Galactica(≈55k) ‡ | 25.61 | 42.78 | 39.53 | 66.03 | 46.25 | 59.34 | 42.61 | 60.34 |
| **RoBERTa** | | | 27.44 | 44.75 | 42.35 | 69.07 | 52.12 | 65.31 | 47.61 | 65.22 |
| **BERT** | ≈11 hrs | 34.3 MB/ Galactica(≈55k) ‡♣ | 25.91 | 43.4 | 40 | 66.99 | 44.63 | 58.34 | 39.57 | 57.87 |
| **RoBERTa** | | | 26.68 | 44.85 | 41.18 | 69.23 | 49.84 | 63 | 47.39 | 64.96 |
| **BERT** | ≈22 hrs | 120.8 MB/ Galactica(≈100k) †‡ | 26.22 | 43.55 | 40.47 | 67.22 | 46.09 | 59.52 | 41.3 | 59.23 |
| **RoBERTa** | | | 28.2 | 45.68 | 43.53 | 70.5 | 52.12 | 65.03 | 49.13 | 66.36 |
| **BERT** | ≈22 hrs | 115.6 MB/ Galactica(≈100k) † ‡ ♣ | 24.23 | 43.27 | 37.41 | 66.79 | 47.07 | 60.79 | 43.26 | 61.58 |
| **RoBERTa** | | | 26.83 | 44.32 | 41.41 | 68.4 | 50.81 | 65.04 | 47.17 | 66.16 |

Table 3: Benchmarking Bio Models (COVID-QA). *: ("18% papers from the computer science domain and 82% from the broad biomedical domain" (Beltagy et al., 2019)); [1](Peng et al., 2019); [2](Yuan et al., 2022); [3](Yamada et al., 2020); Blue/red indicates best/worst scores; **bold = best decoder** (underneath dotted line) scores

| Model | Pre-Training Corpus | Corpus Size | EM | F1 |
|---|---|---|---|---|
| **BioBERT** | PubMed | 4.5B words | 38.14 | 65.65 |
| **SciBERT** | Semantic Scholar* | 3.2B words | 37.99 | 65.96 |
| **SciBERT(+CORD-19)** | Semantic Scholar + CORD-19 | 3.2B words + 20GB | 35.61 | 63.52 |
| **PubMedBERT** | PubMed | 3.1B words / 21GB | 39.03 | 68.56 |
| **BlueBERT** [1] | PubMed + MIMIC | 4.5B words | 29.07 | 56.57 |
| **CODER** [2] | Unified Medical Language System | NA | 38.88 | 66.89 |
| **LUKE** [3] | Wikipedia | 3.5B words | 41.36 | 68.99 |
| **XLNET** | BooksCorpus + Wikipedia + Giga5 + ClueWeb 2012-B + Common Crawl | 32.89B words | 2.38 | 8.83 |
| **Galactica** | c.f. section 2 | 106B tokens | 0 (0) | 5.01 (11.11) |
| **MedLLaMA** | Medical Corpora | NA | 0 (0) | **5.81 (12.79)** |
| **MedAlpaca** | Medical Meadow | NA | **0.03 (0.2)** | 5.21 (12.73) |

performance gap between Clinical and RadBERT. Although Clinical was trained on more clinical data, it was not the *right* data for this task involving radiology report-style documents, leading RadBERT to outperform Clinical on all measures in both splits.

### 4.1.3 Decoder-Based Models

The last 3 rows of Tables 1 & 3 provide zero-shot performance of our chosen decoder models on RadQA & COVID-QA resp. As can be seen, their performance is nowhere near their bidirectional counterparts. Granted they were not fine-tuned, their size, pre-training data coverage & reported performance on related datasets, should have allowed them to at least perform on par or better than open-domain BERT/RoBERTa. **Overall, we see that MedAlpaca seems to be the "best" among** the three for RadQA and only marginally poorer in terms of F1 for COVID-QA. In terms of EM (for COVID-QA), none of the models generated text in line with the gold standard (and hence ~0 EM) and only showed positive F1.

## 4.2 Proposed Method Analysis

### 4.2.1 COVID-QA

**Fine-tuning on our Wikipedia corpus does not yield gains for BERT**, rather a decline of 1.6% in EM, while RoBERTa shows a 2.7% increase in EM and a 0.7% increase in F1. This confirms our hypothesis that **having the right content alone is insufficient without proper structure/style**. However, with our **targeted pre-training, both models demonstrate improvements**. BERT achieves a 5.5% increase in EM and a 2.9% increase in F1,

Table 4: Targeted Pre-training (COVID-QA). Time #: to generate corpus; ⋆: filtered; Gal = Galactica; max_length = Context Max Length. Blue/red indicates best/worst scores.

| Model | Train Dataset | Time# | Corpus Size | EM | F1 |
|---|---|---|---|---|---|
| **BERT** **RoBERTa** | NA [Vanilla Fine-Tuning] | NA | NA | 34.13 39.42 | 60.81 67.5 |
| **BERT** **RoBERTa** | Wikipedia | ≈2.5 hrs | 139.6 MB | 33.58 40.47 | 61.06 67.96 |
| **BERT** **RoBERTa** | Gal(47k) | ≈ 6.5 hrs | 67.4 MB | **36.01** **42.05** | **62.58** **69.3** |
| **BERT** **RoBERTa** | Gal(470k) [10x] | ≈ 2.5 days | 558.2 MB | 34.72 42.2 | 61.39 69.15 |
| **BERT** **RoBERTa** | Gal(25k*2 = 50k)⋆ | ≈ 6.5 hrs | 64.0 MB | 36.45 41.36 | 61.86 69.6 |
| **BERT** **RoBERTa** | Gal(47k) [max_length = 1k] | ≈ 2.5 hrs | 44.8 MB | 34.82 41.46 | 61.11 69.08 |

while RoBERTa shows a 6.7% increase in EM and a 2.7% increase in F1, setting a new SOTA on COVID-QA. Remarkably, RoBERTa even outperforms the previous SOTA model (LUKE) by 1.7% in EM and 0.4% in F1, despite using a training corpus significantly smaller (67.4 MB/0.032B words) than LUKE's 3.5B-word corpus (0.9% of the size).

Contrary to our expectations, **training with a 10x corpus (10 contexts per entity) did not lead to improvements**. Instead, it resulted in minimal enhancements for RoBERTa and even negatively impacted BERT's performance compared to the regular corpus. We attribute this behavior to noise introduced at scale, including ill-formed entities and incorrect facts. As there is currently no reliable method for automatically verifying the integrity of information at scale, we attribute these results to the presence of such noise.

Although **we expected that removing ill-formed entities would improve the results, the fifth row of Table 4 shows that performance actually declined** when we filtered out such entities. We hypothesize that our regular expression-based filtering rules may have mistakenly removed important entities such as author names or URLs, leading to the decline. Furthermore, when we decreased the context length due to limitations in Galactica's token generation (last row of Table 4), we observed a decline in performance for both metrics and both models. This outcome was expected as Galactica was unable to generate content that matched the style of COVID-QA, underscoring the importance of writing style for domain awareness.

### 4.2.2 RadQA

We analyze the results of RadQA separately for each model, considering the type of contexts (prompts) they were trained on and whether they used the filtered or unfiltered set of entities. The "normal" prompt is denoted by "[entity]," while the other prompt is referred to as the "fancy" prompt (see sec. 2). We observed higher test scores on average compared to validation scores, which we attribute to fewer unanswerable questions in the test set (154 vs. 231) and slightly shorter contexts (73.82 vs. 78.1 tokens). We also conducted checks for information leakage but found no irregularities. While we report scores for both sets, our analysis mainly focuses on the dev set, which serves as the first point of evaluation in the RadQA domain.

**When trained on the Wikipedia corpus, BERT shows a decrease in performance on the dev set**, but a 2.5% improvement in EM and a 2.2% improvement in F1 on the test set (versus regular fine-tuning). Training on the unfiltered corpus with normal prompts leads to either a decline or no significant change compared to vanilla fine-tuning and Wikipedia training. This decline or lack of improvement is attributed to noise from ill-formed entities, which were absent from the Wikipedia dataset. However, when the filter is applied, slight improvements are observed over the Wiki corpus (row 2 & 4), particularly in EM (row 1 & 4) for the vanilla baseline. The most **notable improvement for BERT occurs when both filtered entities and the corpus from the fancy prompt is used (row 6)**, resulting in enhancements across all metrics over basic fine-tuning and the Wikipedia baseline (4.3% EM, 0.1% F1 for answerable and overall metrics in basic fine-tuning, and 4.9% EM, 0.6% F1 for answerable and overall metrics in the Wikipedia baseline). It is noteworthy that BERT achieves these scores with a modest 34.3MB corpus, which is << than its benchmarked counterparts.

**RoBERTa** demonstrates improvements across

different combinations of filtration methods and prompt styles, as well as when trained on the Wikipedia corpus. However, the **improvements are less consistent compared to a specific approach**. In terms of **EM**, the **best performance is observed with the corpus using unfiltered entities & normal prompting (row 3)**, with a 4.6% increase over vanilla fine-tuning and a 2.3% increase over the Wikipedia baseline. Regarding **F1, training on the filtered corpus with fancy prompts (row 6) yields the highest increase** of 1.3% over vanilla fine-tuning, but a slight decrease of 0.9% compared to the Wikipedia baseline. Notably, **RoBERTa in row 3 outperforms BioBERT, SciBERT, and LUKE in all metrics**. This is intriguing considering that LUKE is an open-domain model, while the former two are not. Specifically, compared to BioBERT, RoBERTa achieves a 2.3% increase in EM and a 0.9% increase in F1, highlighting the benefits of our approach for domain and dataset awareness.

**We examined the effectiveness of combining different context styles (rows 7 and 8) for our approach**. We created corpora by merging the contexts from both prompt styles for the filtered and the unfiltered entities separately. **The models trained on the unfiltered combined corpora (row 7) showed the best overall performance**. BERT demonstrated a 5.5% increase in EM over regular fine-tuning, a 6.2% increase over the Wikipedia baseline, and similar improvements in F1 by 0.5% and 0.9%, respectively. **RoBERTa** exhibited a 6.9% and 4.5% increase in EM, and a 3.2% and 0.9% increase in F1 compared to their respective baselines, similar to BERT. Moreover, this variant **outperformed ClinicalBERT in F1 by ~2.6% (with roughly the same EM) in addition to surpassing Bio/Sci-BERT, and LUKE. These findings suggest that incorporating a mixture of prompt styles creates a more diverse corpus, enhancing model alignment with the domain. Further, such improvements are achieved with a dataset $<<$ than their bio-based counterparts**.

### 4.3 Investigating Information Leakage

Given that the synthetic corpus generated for COVID-QA in §3 contains entities identified in the *entire* COVID-QA dataset - not from the *train* split within each *fold* - we explore if the performance gains from targeted pre-training are a result of information leak. To this end, we construct a roughly

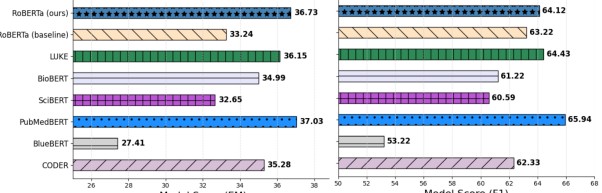

Figure 2: Information Leakage Validation Trials (Left - EM | Right - F1): RoBERTa (ours) was targeted trained on a subset of the 47k corpus with entities only from the 80% train set. All of the other models were fine-tuned in the usual manner i.e. SQuAD→COVID-QA (80% train set) and evaluated on the 20% test set.

80%/20% train/test split (1,676/343 records), ensuring no context overlap, and apply a suite of models to this new split. When applying our targeted pre-training, a syhthetic corpus is generated *only* from entites identified in the train split. The results from a brief parameter search for this assay are presented in Figure 2.

As we can see, the RoBERTa model subjected to targeted pre-training still yields strong performance in this restricted scenario, only surpassed by PubMedBERT (& marginally by LUKE in F1), demonstrating that the **improved performance on COVID-QA cannot be attributed to information leak from the test set**. Although the scores are lower than those in Table 4, the relative scores produced by each model leads to a similar conclusion that targeted pre-training yields optimal results.

## 5   Conclusion & Future Work

We demonstrated the effectiveness of bootstrapping corpora for domain adaptation using FMs, prompting & domain awareness. We achieved SOTA on COVID-QA and observed notable improvements on RadQA by using combinations of corpora, occasionally surpassing the benchmarks. However, this work is just the initial step, and there is room for further exploration. Our future endeavors involve using larger versions of Galactica to enable CoT prompting and to generate even more extensive contexts. Additionally, we aim to incorporate fact-checking mechanisms to eliminate inaccurate information, potentially enhancing the performance of our 10x COVID-QA corpus (c.f. sec. 3). Lastly, beyond corpus, we aspire to explore complete EQA dataset generation that can be used for additional fine-tuning instead of relying solely on pre-training.

## Limitations

We identify two limitations of our work. First, we use a number of GPUs to generate our corpus. While we were fortunate to have access to powerful computing clusters, this could form a bottleneck when being deployed on low-end hardware. However, with cloud services being made more and more affordable, we feel that this point can only be a deal-breaker in severely budget-constrained settings. And second, in this study, we have only shown how to generate corpora for the biomedical domain. For an even wider applicability, we need to study generation techniques for other closed domains such as Finance, Law, Aviation, etc.

## Ethics Statement

As our work relied on publicly available datasets, we believe that the ethical ramifications here are limited. That being said, we recognize that to use RadQA, we had to acquire certifications to access it. This shows that even though the data in it is redacted, loosely disseminated patient reports are a threat to their privacy. Moreover, we had to make sure that when generating our synthetic reports, we were not mentioning any patient names, which even with a small probability might bear resemblance to an actual person.

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

## A   Model Cards

All models used in this study were downloaded from the HuggingFace library (Wolf et al., 2020). Each model, along with its model card (name as it appears in the HuggingFace model hub) and URL is listed in Table 5.

## B   Hardware Details

To run our corpus generation code, we used a total of 8 40GB NVIDIA A100 or 8 32GB NVIDIA Tesla V100 cards, depending on our institutes job scheduler and availability. Although the former is a superior card, we preferred the latter since it gives relatively quick turnaround and is a general purpose GPU unlike the A100 which is required for even heavier compute. In fact we were requested by our system administrator to use the V100 over the A100, if it was possible, to avoid blocking off even more compute-heavy experiments, which we did not have an issue obeying. That being said, if the A100 cluster was free, we did take advantage of it to run our code. For pre-training and fine-tuning BERT/RoBERTa, we only used 1 GPU, out of either cluster.

## C   Hyperparameters Used

Hyperparameters for each experiment is detailed in Table 6. These were selected mostly from preexisting implementations or through minimal exploration of known settings.

## D   Synthetic Corpora Samples

We provide samples from our generated corpus. In Figure 3 we show two positive results for our COVID-QA directed corpus. The top one is in the style of a research paper while the bottom one, though shorter in length, details useful information on the required entity. We show negative examples of the same in Figure 4. The top one completely degrades into noise while the bottom one although coherent is talking about an unrelated topic.

For RadQA, we show similar examples. In Figure 6 we show positive (top) and negative (bottom) samples from normal prompting. Note that since programming languages were a part of Galactica's training corpora, it sometimes generates the same during prompting. Finally, Figure 7 shows positive (top) and negative (bottom) examples of fancy prompting. While the positive sample shows a strong correlation with an actual radiology report, the bottom one completely breaks down.

## E   Prompts Tried

As mentioned before, we knew the prompt to be used for COVID-QA since it was a feature of Galactica. However, we had to experiment with a few prompt styles for RadQA before landing on our final one. We describe the prompts we tried below for the entity `small-bowel injury` and the reasons for not using them. The text in `courier` is the prompt and in *italics* is the generated text.

- IMPRESSION: small-bowel injury *(SBI) was defined as injury to the small bowel wall, or transmural disruption of the bowel wall, in the absence of penetrating abdominal trauma ( The management of small-bowel injuries in the emergency room., Brown). Patients were excluded if the SBIs occurred after admission to the emergency department or were complicated by perforation.* ... (truncated)

    - Issue - Did not include the `FINDINGS` section.

**Title: chest pain** *Title: an unusual cause of chest pain in an elderly patient*
*Authors: Sushila Shrivastava, Vinay Kumar, Pramod Kumar*
*Keywords: Chest pain, Eosinophilic pneumonia, Chest pain*

*# Abstract*

*# Background*

*Pulmonary eosinophilic infiltration is one of the rare causes of non-specific chest pain in the elderly patients. We present a case of elderly patient who presented with a non-specific chest pain of 1 day duration with eosinophilia and chest CT scan showing ground glass opacities.*

*# Case presentation*

*A 74-year-old lady presented with a 1-day history of chest pain, nausea and vomiting. The patient had a history of cough for 1 year, and she was on regular medication for cough. The patient had a significant history of hypertension, diabetes mellitus and COPD. On examination, the patient was afebrile with an oxygen saturation of 98% on room air. She had a heart rate of 90 beats per minute and blood pressure of 115/70 mmHg. Chest auscultation revealed a crackles in the lower lung fields. Her chest X-ray showed bilateral pulmonary infiltrates. A diagnosis of eosinophilic pneumonia was made and she was started on oral corticosteroids. She recovered without any further complications after two weeks of treatment.*

*# Conclusion*

*Chest pain in the elderly patient is not always indicative of a life-threatening condition. A high index of suspicion for this rare entity should be kept in mind when a non-specific chest pain in elderly patients is seen.*

*# Background*

*Pulmonary eosinophilic infiltration is a rare cause of non-specific chest pain in elderly patients. The clinical presentation of the disease is similar to other lung diseases with non-specific symptoms, and a high index of suspicion for this rare entity is required in elderly patients presenting with non-specific chest pain. We present a case of a 74-year-old lady with a history of cough who presented with a non-specific chest pain of 1 day duration with eosinophilia.*

*# Case presentation*

*A 74-year-old lady was admitted in a state of mild depression. The patient had a significant history of hypertension, diabetes mellitus and COPD. She had a history of chronic cough for 1 year and was on regular medication for the cough. On examination, the patient was afebrile with an oxygen saturation of 98% on room air. She had a heart rate of 90 beats per minute and blood pressure of 115/70 mmHg. Chest auscultation revealed a crackles in the lower lung fields. She was investigated with a chest X-ray which showed bilateral pulmonary infiltrates. An ECG showed no sign of ST-T segment elevation. Her complete blood count (CBC) was as follows: haemoglobin 12.4 gm%, leucocytes 10,300/cu mm, platelets 253,000/cu mm, red blood cells (RBCs) 2.28 million/cu mm, and haemoglobin percentage (Hb%) 24.9%. Her electrolytes were normal. Her C-reactive protein (CRP) was 37.1 mg/L, D-dimers 0.24 μg/ml, and erythrocyte sedimentation rate (ESR) was 66 mm/1st hour. Her sputum was clear. The chest X-ray showed bilateral pulmonary infiltrates (Fig. 1). A diagnosis of eosinophilic pneumonia was made, and she was started on oral corticosteroids. She recovered without any further complications after two weeks of treatment.Fig. 1: Chest X-ray shows bilateral pulmonary infiltrates*

*# Discussion*

*Eosinophilic pneumonia is an uncommon cause of non-specific chest pain. The incidence of this condition is approximately 0.1% of all pulmonary infiltrations [ Idiopathic eosinophilic pneumonia: clinical features, radiological appearances and long-term outcome., Brown, Eosinophilic pneumonia: imaging findings., Bongiovanni]. The most common age group affected is between 40 and 60 years [ Idiopathic eosinophilic pneumonia: clinical features, radiological appearances and long-term outcome., Brown]. However, there have been several case reports of eosinophilic pneumonia in the elderly, with the peak age between 60 and 70 years [ Eosinophilic pneumonia in elderly patients: a case series and review of the literature, Kara, Eosinophilic pneumonia in the elderly, De Luca]. Most of the patients with eosinophilic pneumonia are asymptomatic at presentation, but it can also present with fever, dyspnea, cough, and chest pain [ Idiopathic eosinophilic pneumonia: clinical features, radiological appearances and long-term outcome., Brown, Eosinophilic pneumonia in elderly patients: a case series and review of the literature, Kara, Eosinophilic pneumonia in the elderly, De Luca]. The chest pain in patients with eosinophilic pneumonia can be localized or generalized and can be of mild, moderate or severe intensity. It can be persistent or acute in nature [ Idiopathic eosinophilic pneumonia: clinical features, radiological appearances and long-term outcome., Brown, Eosinophilic pneumonia in the elderly, De Luca]. The patient can be admitted to hospital with a cough, dyspnea, chest pain and fever. However, the patient may not show any of these symptoms at presentation [ Idiopathic eosinophilic pneumonia: clinical features, radiological appearances and long-term outcome., Brown, Eosinophilic pneumonia in the elderly, De Luca].*

*The most common clinical feature of eosinophilic pneumonia is cough, which is present in 85% of patients [ Idiopathic eosinophilic pneumonia: clinical features, radiological appearances and long-term outcome., Brown]. Fever is present in 50% to 75% of patients [ Idiopathic eosinophilic pneumonia: clinical features, radiological appearances and long-term outcome., Brown, Eosinophilic pneumonia in the elderly, De Luca]. Dyspnea and chest pain are seen in less than 10% of patients [ Idiopathic eosinophilic pneumonia: clinical features, radiological appearances and long-term outcome., Brown, Eosinophilic pneumonia in the elderly, De Luca]. The most common respiratory symptoms are cough and dyspnea [ Eosinophilic pneumonia in the elderly, De Luca]. The majority of the patients with eosinophilic pneumonia are asymptomatic, while a minority of patients present with non-specific symptoms of cough, fever, dyspnea and chest pain [ Idiopathic eosinophilic pneumonia: clinical features, radiological appearances and long-term outcome., Brown, Eosinophilic pneumonia in the elderly, De Luca]. Patients with pulmonary eosinophilia and elevated ESR or CRP have been reported to have a poorer prognosis [ Idiopathic eosinophilic pneumonia: clinical features, radiological appearances and long-term outcome., Brown]. Chest radiography can be nonspecific and shows bilateral or unilateral infiltrates [ Idiopathic eosinophilic pneumonia: clinical features, radiological appearances and long-term outcome., Brown, Eosinophilic pneumonia in the elderly, De Luca]. A chest CT scan is often required to diagnose eosinophilic pneumonia and show a variety of appearances. These include ground glass opacities (GGO) and reticular opacities [ Idiopathic eosinophilic pneumonia: clinical features, radiological appearances and long-term outcome., Brown, Eosinophilic pneumonia in the elderly, De Luca]. In our case, the patient had a history of chronic cough and had no fever on presentation. A chest X-ray showed bilateral infiltrates. A chest CT scan showed GGO and reticular opacities.*

*The etiology of eosinophilic pneumonia is unknown, but it has been proposed that it is a T-helper 2-mediated inflammatory disorder [ Idiopathic eosinophilic pneumonia: clinical features, radiological appearances and long-term outcome., Brown]. The diagnosis of eosinophilic pneumonia can be difficult because of the absence of typical radiological findings. It can be difficult to distinguish eosinophilic pneumonia from other inflammatory lung diseases. Treatment consists of corticosteroids, and the use of immunosuppressive drugs, such as azathioprine, hydroxychloroquine, and methotrexate, has been reported to improve the symptoms of eosinophilic pneumonia [ Idiopathic eosinophilic pneumonia: clinical features, radiological appearances and long-term outcome., Brown, Eosinophilic pneumonia in the elderly, De Luca].*

*# Conclusion*

*Eosinophilic pneumonia is a rare cause of non-specific chest pain in elderly patients. In these patients, the diagnosis can be difficult and a high index of suspicion should be kept in mind in such patients. Early diagnosis and prompt treatment can improve the prognosis.*

*# Consent*

*Written informed consent was obtained from the patient for publication of this case report and any accompanying images. A copy of the written consent is available for review by the Editor-in-Chief of this journal.*

*Footnote: Competing interests*

*The authors declare that they have no competing interests.*

*Footnote: Authors' contributions*

*SS, KK and PK were involved in the management of the patient. SS drafted the manuscript. All authors read and approved the final manuscript.*

- - - - - - - - - - - - - - - - - - - - - - - - - - - - - - - - - - - - - - - - - - - - - - - - - - - - - - - - - - - - - - - - - - - -

**Title: Polymerase chain reaction** *#1: a new diagnostic tool in the management of tuberculosis.*

*Abstract: Polymerase chain reaction (PCR) is an advanced molecular diagnostic technique. The potential of PCR in the diagnosis of tuberculosis (TB) has been evaluated by several investigators. PCR techniques have been used in the diagnosis of extra-pulmonary TB as well as the diagnosis of drug resistance. This article reviews the current status of PCR in the diagnosis of TB.*

Figure 3: **COVID-QA positive** examples (underline = entity | red = prompt) | olive = generation

**Title: virion perimetrical** #1032; #1034; #1036; #1038; #1040; #1042; #1044; #1046; #1048; #1050; #1052; #1054; #1056; #1058; #1060; #1062; #1064; #1066; #1068; #1070; #1072; #1074; #1076; #1078; #1080; #1082; #1084; #1086; #1088; #1090; #1092; #1094; #1096; #1098; #1100; #1102; #1104; #1106; #1108; #1110; #1112; #1114; #1116; #1118; #1120; #1122; #1124; #1126; #1128; #1130; #1132; #1134; #1136; #1138; #1140; #1142; #1144; #1146; #1148; #1150; #1152; #1154; #1156; #1158; #1160; #1162; #1164; #1166; #1168; #1170; #1172; #1174; #1176; #1178; #1180; #1182; #1184; #1186; #1188; #1190; #1192; #1194; #1196; #1198; #1200; #1202; #1204; #1206; #1208; #1210; #1212; #1214; #1216; #1218; #1220; #1222; #1224; #1226; #1228; #1230; #1232; #1234; #1236; #1238; #1240; #1242; #1244; #1246; #1248; #1250; #1252; #1254; #1256; #1258; #1260; #1262; #1264; #1266; #1268; #1270; #1272; #1274; #1276; #1278; #1280; #1282; #1284; #1286; #1288; #1290; #1292; #1294; #1296; #1298; #1300; #1302; #1304; #1306; #1308; #1310; #1312; #1314; #1316; #1318; #1320; #1322; #1324; #1326; #1328; #1330; #1332; #1334; #1336; #1338; #1340; #1342; #1344; #1346; #1348; #1350; #1352; #1354; #1356; #1358; #1360; #1362; #1364; #1366; #1368; #1370; #1372; #1374; #1376; #1378; #1380; #1382; #1384; #1386; #1388; #1390; #1392; #1394; #1396; #1398; #1400; #1402; #1404; #1406; #1408; #1410; #1412; #1414; #1416; #1418; #1420; #1422; #1424; #1426; #1428; #1430; #1432; #1434; #1436; #1438; #1440; #1442; #1444; #1446; #1448; #1450; #1452; #1454; #1456; #1458; #1460; #1462; #1464; #1466; #1468; #1470; #1472; #1474; #1476; #1478; #1480; #1482; #1484; #1486; #1488; #1490; #1492; #1494; #1496; #1498; #1500; #1502; #1504; #1506; #1508; #1510; #1512; #1514; #1516; #1518; #1520; #1522; #1524; #1526; #1528; #1530; #1532; #1534; #1536; #1538; #1540; #1542; #1544; #1546; #1548; #1550; #1552; #1554; #1556; #1558; #1560; #1562; #1564; #1566; #1568; #1570; #1572; #1574; #1576; #1578; #1580; #1582; #1584; #1586; #1588; #1590; #1592; #1594; #1596; #1598; #1600; #1602; #1604; #1606; #1608; #1610; #1612; #16

**Title: ADA** Title: Automatic Data Augmentation for Text Classification

*Authors: Koushik Ghosh, Kishore Kumar Chaudhuri, Siddharth Jha, Abhishek Kumar*

# Abstract

*A data set is small in size and is not representative of the entire population. So it is important to train a model that can perform well on unseen data, and this can be achieved by using various types of data augmentation techniques. Data augmentation techniques are applied to the data to create synthetic examples. However, most of the existing approaches of data augmentation are heuristic and do not make use of all the available information. In this paper, we propose a general framework to automatically generate the synthetic examples for a given dataset. The proposed framework is based on the observation that the model needs to be trained on the augmented data set for obtaining the better generalization performance. We show that using the proposed framework one can achieve up to $2.4\%$ higher accuracy on a real-world data set with an average of $6.4\%$ more data augmentation.*

# 1 Introduction

*An important step in supervised machine learning is the creation of a labelled data set for training a classifier. In this step, a large set of examples is obtained from an unlabeled data set and manually annotated with a label [ Data Augmentation for Text Classification: A Survey, Zhang]. A labelled data set can be very expensive and time consuming to create. One way to reduce the cost to obtain the annotated data set using data augmentation techniques.*

*Data augmentation techniques are commonly used in image classification to create synthetic examples that are very similar to the original ones [ A survey on Image Data Augmentation for Deep Learning, Shorten, mixup: Beyond Empirical Risk Minimization, Zhang, mixup-v0: Improved Mixup for Data Augmentation, Kumar]. These techniques have also been extended to other data types such as text, audio, video etc. [ Data Augmentation for Text Classification: A Survey, Zhang]. The key idea is to generate synthetic examples which have similar characteristics as the original examples.*

*Data augmentation techniques are often heuristic in nature. They are designed to be intuitive and easy to use, but they do not make use of the available information and hence they often perform sub-optimally. In this paper, we propose a general framework to automatically generate the synthetic examples for a given dataset. The proposed framework is based on the observation that the model needs to be trained on the augmented data set for obtaining the better generalization performance. We show that using the proposed framework one can achieve up to $2.4\%$ higher accuracy on a real-world data set with an average of $6.4\%$ more data augmentation.*

*The main contributions of this work are as follows:*

*1. We propose a general framework to automatically generate the synthetic examples for a given dataset. The proposed framework is based on the observation that the model needs to be trained on the augmented data set for obtaining the better generalization performance.*
*2. We show that using the proposed framework one can achieve up to $2.4\%$ higher accuracy on a real-world data set with an average of $6.4\%$ more data augmentation.*

# II Related work

# II-a Data Augmentation for Image Classification

*There have been many proposed data augmentation techniques for image classification. For example, Cutout [ Improved Regularization of Convolutional Neural Networks with Cutout, Devries] and Cutmix [ CutMix: Regularization Strategy to Train Strong Classifiers With Localizable Features, Yun] techniques are proposed to cut out a portion of the image and paste it on the other part of the image in an random way. Data augmentation techniques proposed for text classification are also quite diverse. Examples of such techniques include: random erasing [ Random Erasing Data Augmentation, Zhong], semantic augmentation [ Semantic Data Augmentation for Deep Learning, Tao], random re-ordering of words [ Text Augmentation for Learning Natural Language Inference Models, Zhang], random word deletion [ Data Augmentation for Low-Resource Neural Machine Translation, Fadaee], etc. A detailed survey on image data augmentation techniques can be found in [ A survey on Image Data Augmentation for Deep Learning, Shorten].*

# II-B Data Augmentation for Text Classification

*Data augmentation techniques are very common in text classification tasks. The techniques proposed for text classification can be broadly divided into two categories: generative data augmentation and heuristic data augmentation. Generative data augmentation methods generate synthetic examples by training a generator model. For example, text completion [ A Hybrid Method for Text Classification with Generative Adversarial Network and Deep Learning, Jahan] and word dropout [ Data Augmentation for Low-Resource Neural Machine Translation, Fadaee] generate synthetic examples by training a generator model. Heuristic data augmentation techniques generate synthetic examples by performing some simple operations on the original examples. For example, Mixup [ mixup: Beyond Empirical Risk Minimization, Zhang], Cutmix [ CutMix: Regularization Strategy to Train Strong Classifiers With Localizable Features, Yun] and Cutout [ Improved Regularization of Convolutional Neural Networks with Cutout, Devries] use randomly selected examples from the original examples and add them together.*

*Generative data augmentation techniques have been extended to other data types such as images, audio, video etc. [ Data Augmentation for Text Classification: A Survey, Zhang]. However, the heuristic data augmentation techniques are used in the majority of the text classification tasks. Examples of heuristic data augmentation techniques for text classification include: the random swapping of two words [ Text Augmentation for Learning Natural Language Inference Models, Zhang], the random insertion of a random word at random position in a sentence [ Text Augmentation for Learning Natural Language Inference Models, Zhang], the random word deletion [ Data Augmentation for Low-Resource Neural Machine Translation, Fadaee], etc.*

# III Problem Formulation

*We consider a given data set $X=\{x_{1}, x_{2}, \ldots x_{N}\}$ where $x_{i} \in \mathbb{R}^{d}$ is the $i^{th}$ example with $d$ dimension. The objective of supervised learning is to train a model $f_{\theta}$ for a given classification task. The model $f_{\theta}$ takes a set of input examples $X$ and outputs a prediction of the class for each input example. We denote the set of labels for the data set as $Y=\{y_{1}, y_{2}, \ldots, y_{N}\}$ where $y_{i}$ is the label for the $i^{th}$ example. The training set is composed of a set of pairs $S=\{(x_{i}, y_{i}), i=1\}^{N}$, where $(x_{i}, y_{i})$ is the pair for the $i^{th}$ example. The training objective is to find the model parameters $\theta^{*}$ that minimizes the training error.*

*The data augmentation technique, $T_{aug}$, is used to generate a set of augmented examples, $X'=\{x_{1}', x_{2}', \ldots, x_{N}'\}$, from the original data set, $X$. The generated data set $X'$ has similar characteristics as the original set $X$.*

*The training objective is to find the model parameters $\theta^{*}$ that minimizes the training error, $e_{train}(\theta)$, using the augmented set $X'$. We denote the model trained using the augmented set $X'$ as $f_{\theta}'$.*

# IV Proposed Framework

*Fig. 1: Framework of the proposed framework*

*We propose a framework for automatic data augmentation which is shown in Fig. 1. The proposed framework is based on the following observations: 1) the model needs to be trained on the augmented data set for obtaining the better generalization performance, and 2) using the proposed framework one can achieve up to $2.4\%$ higher accuracy on a real-world data set with an average of $6.4\%$ more data augmentation.*

*We use the concept of domain adaptation in our framework. This idea is based on the observation that a model needs to be trained on the augmented data set for obtaining the better generalization performance. Let $\mathcal{L}_{train}$ be the training loss function, $\mathcal{D}$ be the domain, $T_{aug}$ be the data augmentation technique, $X'$ be the set of*

Figure 4: **COVID-QA negative** examples (underline = entity | red = prompt) | olive = generation

---

**Question:** What is DC-GENR and where is it expressed?
**ChatGPT:** *As an AI language model, I do not have access to up-to-date information or research beyond my training data. Therefore, I am not aware of any current information on "DC-GENR." It is possible that this refers to a specific term or concept within a particular field or context. If you could provide me with more context or information about what you are referring to, I may be able to assist you further.*

**Question:** What is the main cause of HIV-1 infection in children?
**ChatGPT:** *The main cause of HIV-1 infection in children is mother-to-child transmission of the virus during pregnancy, childbirth, or breastfeeding. This is also known as vertical transmission. If the mother is HIV-positive and not receiving treatment, there is a high risk of transmitting the virus to her baby. However, with proper medical care and treatment, the risk of mother-to-child transmission can be significantly reduced.*

**Question:** What plays the crucial role in the Mother to Child Transmission of HIV-1 and what increases the risk?
**ChatGPT:** *The level of the mother's HIV viral load plays a crucial role in mother-to-child transmission of HIV-1. A higher viral load in the mother's blood and genital secretions increases the risk of transmission to the baby. Factors that can increase the risk of mother-to-child transmission include lack of antiretroviral therapy during pregnancy, vaginal delivery, and breastfeeding. However, with proper medical care and treatment, the risk of mother-to-child transmission can be greatly reduced. Pregnant women living with HIV should seek medical care and treatment to help protect their babies*

Figure 5: Testing ChatGPT

```
displacement WhatIs()) {
    EXPECT_EQ(1, dynamic_cast<const Shape&>(shape_->shape()).dimensions());
    EXPECT_EQ(1, shape_->rank());
    EXPECT_TRUE(IsSameRank(shape_->shape(), shape_));
    EXPECT_EQ(1, shape_->dimensions());
  }
}

TEST_P(ShapeOpTest, ShapeNotSameRank_1D) {
  Shape shape_;
  shape_.set_rank(1);
  shape_.set_dims({1});
  Shape shape_copy = *shape_;
  shape_.set_rank(0);
  shape_copy.set_rank(1);
  Shape expected(shape_copy);
  TF_EXPECT_OK(
      shape_.AddScalar(2.0, /*add_shape=*/nullptr, /*is_dynamic=*/false));
  TF_EXPECT_OK(shape_.AddScalar(3.0, /*add_shape=*/nullptr, /*is_dynamic=*/false));
  TF_EXPECT_OK(shape_.AddScalar(4.0, /*add_shape=*/nullptr, /*is_dynamic=*/false));
  TF_EXPECT_OK(
      shape_.AddScalar(2.0, /*add_shape=*/nullptr, /*is_dynamic=*/true));
  TF_EXPECT_OK(shape_.AddScalar(3.0, /*add_shape=*/nullptr, /*is_dynamic=*/true));
  TF_EXPECT_OK(shape_.AddScalar(4.0, /*add_shape=*/nullptr, /*is_dynamic=*/true));
  TF_EXPECT_OK(shape_.AddScalar(2.0, /*add_shape=*/nullptr, /*is_dynamic=*/true));
  TF_EXPECT_OK(shape_.AddScalar(3.0, /*add_shape=*/nullptr, /*is_dynamic=*/true));
  TF_EXPECT_OK(shape_.AddScalar(4.0, /*add_shape=*/nullptr, /*is_dynamic=*/true));
  TF_EXPECT_OK(shape_.AddScalar(2.0, /*add_shape=*/nullptr, /*is_dynamic=*/true));
  TF_EXPECT_OK(shape_.AddScalar(3.0, /*add_shape=*/nullptr, /*is_dynamic=*/true));
  TF_EXPECT_OK(shape_.AddScalar(4.0, /*add_shape=*/nullptr, /*is_dynamic=*/true));
  TF_EXPECT_OK(shape_.AddScalar(2.0, /*add_shape=*/nullptr, /*is_dynamic=*/true));
  TF_EXPECT_OK(shape_.AddScalar(3.0, /*add_shape=*/nullptr, /*is_dynamic=*/true));
  TF_EXPECT_OK(shape_.AddScalar(4.0, /*add_shape=*/nullptr, /*is_dynamic=*/true));

  EXPECT_TRUE(ShapeIsSame(shape_, expected));
}

TEST_P(ShapeOpTest, ShapeNotSameRank_2D) {
  Shape shape_;
  shape_.set_rank(1);
  shape_.set_dims({1, 2});
  shape_.set_strides({1, 2});
  shape_.set_sizes({1, 2});
  Shape shape_copy = *shape_;
  shape_.set_rank(0);
  shape_copy.set_rank(1);
  shape_copy.set_dims({1, 2});
  shape_copy.set_strides({1, 2});
  shape_copy.set_sizes({1, 2});
  Shape expected(shape_copy);
  TF_EXPECT_OK(
      shape_.AddScalar(2.0, /*add_shape=*/nullptr, /*is_dynamic=*/false));
  TF_EXPECT_OK(shape_.AddScalar(3.0, /*add_shape=*/nullptr, /*is_dynamic=*/false));
  TF_EXPECT_OK(shape_.AddScalar(4.0, /*add_shape=*/nullptr, /*is_dynamic=*/false));
  TF_EXPECT_OK(
      shape_.AddScalar(2.0, /*add_shape=*/nullptr, /*is_dynamic=*/true));
  TF_EXPECT_OK(shape_.AddScalar(3.0, /*add_shape=*/nullptr, /*is_dynamic=*/true));
  TF_EXPECT_OK(shape_.AddScalar(4.0, /*add_shape=*/nullptr, /*is_dynamic=*/true));
  TF_EXPECT_OK(shape_.AddScalar(2.0, /*add_shape=*/nullptr, /*is_dynamic=*/true));
  TF_EXPECT_OK(shape_.AddScalar(3.0, /*add_shape=*/nullptr, /*is_dynamic=*/true));
  TF_EXPECT_OK(shape_.AddScalar(4.0, /*add_shape=*/nullptr, /*is_dynamic=*/true));

  EXPECT_TRUE(ShapeIsSame(shape_, expected));
}

TEST_P(ShapeOpTest, ShapeNotSameRank_3D) {
  Shape shape_;
  shape_.set_rank(1);
  shape_.set_dims({1, 2, 3});
  shape_.set_strides({1, 2, 3});
  shape_.set_sizes({1, 2, 3});
  Shape shape_copy = *shape_;
  shape_.set_rank(0);
  shape_copy.set_rank(1);
  shape_copy.set_dims({1, 2, 3});
  shape_copy.set_strides({1, 2, 3});
  shape_copy.set_sizes({1, 2, 3});
  Shape expected(shape_copy);
  TF_EXPECT_OK(
      shape_.AddScalar(2.0, /*add_shape=*/nullptr, /*is_dynamic=*/false));
  TF_EXPECT_OK(shape_.AddScalar(3.0, /*add_shape=*/nullptr, /*is_dynamic=*/false));
  TF_EXPECT_OK(shape_.AddScalar(4.0, /*add_shape=*/nullptr, /*is_dynamic=*/false));
  TF_EXPECT_OK(
      shape_.AddScalar(2.0, /*add_shape=*/nullptr, /*is_dynamic=*/true));
  TF_EXPECT_OK(shape_.AddScalar(3.0, /*add_shape=*/nullptr, /*is_dynamic=*/true));
  TF_EXPECT_OK(shape_.AddScalar(4.0, /*add_shape=*/nullptr, /*is_dynamic=*/true));
  TF_EXPECT_OK(shape_.AddScalar(2.0, /*add_shape=*/nullptr, /*is_dynamic=*/true));
  TF_EXPECT_OK(shape_.AddScalar(3.0,
```

Figure 6: RadQA - normal prompt (underline = entity | red = prompt) | olive = generation

| Model | Model Card (URL) |
|---|---|
| BERT-Base, Cased | bert-base-cased |
| BERT-Base, Cased, SQuAD v1 | batterydata/bert-base-cased-squad-v1 |
| BERT-Base, Cased, SQuAD v2 | deepset/bert-base-cased-squad2 |
| RoBERTa-Base | roberta-base |
| RoBERTa-Base, SQuAD v1 | csarron/roberta-base-squad-v1 |
| RoBERTa-Base, SQuAD v2 | deepset/roberta-base-squad2 |
| BioBERT | dmis-lab/biobert-base-cased-v1.2 |
| SciBERT | allenai/scibert_scivocab_uncased |
| SciBERT (+CORD-19) | lordtt13/COVID-SciBERT |
| PubMedBERT | microsoft/BiomedNLP-PubMedBERT-base-uncased-abstract-fulltext |
| BlueBERT | bionlp/bluebert_pubmed_mimic_uncased_L-12_H-768_A-12 |
| CODER | GanjinZero/UMLSBert_ENG |
| LUKE | studio-ousia/luke-base |
| XLNet, SQuAD v1 | arrafmousa/xlnet-base-cased-finetuned-squad |
| STonKGs * | stonkgs/stonkgs-150k |
| RadBERT | zzxslp/RadBERT-RoBERTa-4m |
| Clinical BERT | emilyalsentzer/Bio_ClinicalBERT |
| BioMed-RoBERTa | allenai/biomed_roberta_base |
| MedLLaMA | chaoyi-wu/MedLLaMA_13B |
| MedAlpaca | medalpaca/medalpaca-13b |
| Galactica | facebook/galactica-1.3b |

Table 5: Model cards and URLs for all models used in our paper. * We wanted to use STonKGs (Balabin et al., 2022). However, there was no vocabulary file for the model which resulted in errors.

- FINDINGS: small-bowel injury *was observed in 11.1% and 21.9% of patients, respectively. Small-bowel injury associated with liver injury occurred in 10.1%. The mortality rate was significantly higher in the small-bowel injury group than in the non-small-bowel injury group (11.6% vs. 2.1%, P = 0.005). This study shows that small-bowel injury is associated with a high mortality rate, and that this complication is an important cause of morbidity and mortality in patients with severe acute pancreatitis.*

  – Issue - Did not include the IMPRESSIONS section.

- Patient has small-bowel injury. FINAL REPORT *This patient was treated with conservative methods. The small-bowel injury was treated surgically with resection of the bowel, primary anastomosis, and end-to-end anastomosis between the ends of the resected bowel. She did not have recurrence of intestinal obstruction after 3 years.*

  – Issue - Did not include either IMPRESSIONS or FINDINGS section.

- John Doe has small-bowel injury. IMPRESSIONS *: This case study was made possible by his support by a research grant from the National Institutes of Health (CA 10644).*

  – Issue - Had to make sure that the pronouns used were same throughout the report otherwise it would not be consistent.

- Jane Doe has small-bowel injury. IMPRESSIONS *: This case study was made possible by the support of a research grant from the Department of Veterans Affairs.*

  – Issue - Same as above.

- Patient I has small-bowel injury *He has no history of abdominal trauma or surgery. At first, he has only mild abdominal pain. But with the progress of disease, he has become distressed. At the initial examination, the abdominal pain was alleviated, but the patient still has slight*

Figure 7: RadQA - fancy prompt (underline = entity | red = prompt) | olive = generation

| Experiment | Hyperparameters |
|---|---|
| Corpus Generation | random seed: 42
renormalize_logits: True
do_sample: True
max_length (prompt + generated tokens): 2,048
top_p: 0.9
temperature: 0.9 |
| Pre-Training | batch_size: 40
learning_rate: 5e-5
epochs: 3 |
| Fine-Tuning (SQuAD) | batch_size: 16
max_input_length (question + context): 384
stride: 128
learning_rate: 2e-5
epochs: 3
n_best (top n answer spans): 20
max_answer_length: 30
optimizer_type: AdamW |
| Fine-Tuning (COVID-QA) | batch_size: 40
max_input_length (question + context): 384
stride: 128
learning_rate: 2e-5
epochs: 1
n_best (top n answer spans): 20
max_answer_length: 1000
optimizer_type: AdamW |
| Fine-Tuning (RadQA) | batch_size: 16
max_length: 384
stride: 128
learning_rate: 3e-5
epochs: 1
n_best (top n answer spans): 20
max_answer_length: 1000
optimizer_type: AdamW |

Table 6: Hyperparameters for each experiment. Note, all random seeds were set to 42. This was especially important for Galactica since a different seed implied different sampling of the vocabulary. Additionally, for reproducibility, the seed was set to 42.

*distention of abdomen. He was admitted to our hospital on the third day of admission. His vital signs were: body temperature 37.5 °C, blood pressure 110/70 mmHg, pulse 105 beats/min, respiratory rate 26 breaths/min and oxygen saturation 97% on room air. He had mild abdominal distention. The results of routine blood tests, urinalysis and abdominal ultrasound examination were normal. X-ray showed free air in the abdomen. ... (truncated)*

- – Issue - Galactica could get confused between I (alphabet) and I (roman numeral).

- `Patient A has small-bowel injury` *(Fig. 1). He has no history of abdominal trauma or surgery. At first, an abdominal CT scan was performed to evaluate abdominal pain. Abdominal CT showed a partial small-bowel obstruction at the splenic flexure of the colon (Fig. 2). An oral contrast medium was then administered via a nasogastric tube and an abdominal CT scan was performed. ... (truncated)*

- – Issue - Galactica could get confused misinterpret "A" for the beginning of a sen-

tence (it was observed for a few cases)

- `Patient alpha has small-bowel injury` *requiring an open operation and was diagnosed with C. difficile colitis. The patient had a history of severe anemia and leukopenia, which were also observed in patient beta. Patient alpha was treated with a single course of intravenous moxifloxacin 400 mg/day, which resulted in the resolution of the diarrhea. He was followed closely after discharge. In patient beta, he had a similar episode of diarrhea after hospital admission, but the diarrhea did not resolve. He was prescribed with an oral vancomycin 1000 mg twice a day, and the diarrhea was resolved. ...* (truncated)

  - Issue - This would have been a good choice. However, as we see above, patient qualifiers such as `A`, `I` and `alpha`, *might* bias the model towards more *male patients* (He). As such, we decided to drop the qualifier altogether.

## F ChatGPT results

We were curious to see if ChatGPT (with GPT3.5) would be able to answer a few questions from COVID-QA. Only the questions were provided to it without the context since they were much longer than its input capacity. We did not test RadQA since its questions were heavily linked to the patient report on hand and were not as *general* as the former's. Figure 5 shows a few examples from our trials. As we can see, it seems to answer the questions correctly. However, as mentioned before, these answers need validation from a professional in the field. Also, we can see that sometimes it **cannot answer** questions on topics not found in its training data, a problem not faced by our encoder-only models.

## G Note on Stability

We have noticed that RoBERTa is an extremely stable architecture i.e. all scores from pre-training to both rounds of fine-tuning were consistent, down to the final decimal point, over two consecutive runs. However, BERT was a little less so. We observed that it is more sensitive to pre-training and subsequently gives a slight deviation in its downstream scores. That notwithstanding, we did see that in Table 2, when using the corpus from the filtered entities and fancy prompt, BERT showed RoBERTa-like behavior i.e. consistency in all scores across each phase of training. Overall, we report first-time runs for BERT.

