# OpenReview forum: "Quality > Quantity: Synthetic Corpora from Foundation Models for Closed-Domain Extractive Question Answering"
_EMNLP/2023/Conference — Submitted to EMNLP 2023_

### Official Review · Reviewer_grqj · 2023-08-06

**Soundness:** 3

**Excitement:**

4: Strong: This paper deepens the understanding of some phenomenon or lowers the barriers to an existing research direction.

**Paper Topic And Main Contributions:**

The authors propose targeted pre-training as a solution for a closed-domain extractive QA task. Using a generative LM they generate task focused pre-training data for two biomedical extractive question answering datasets, COVID-QA and RadQA. The pre-trained model achieves a new benchmark on the former and demonstrates overall improvements on the latter.



**Reasons To Accept:**

1. The submission proposes an end-to-end flow for creating an LM for closed-domain tasks. It is well written and well structured.
2. The authors use a generative LM for dataset generation in a closed-domain task where data privacy is a concern. Although numerous recent works make use of LLMs for dataset generation, the authors clearly outline the motivation to do so for the medical extractive QA task pre-training.
3. The work also clearly outlines the motivation for redefining the domain-specific pre-training paradigm and carry out experiments evaluating the same.
4. The authors also carry out extensive ablation studies to evaluate the impact of different prompts, corpus size, context length, and wikipedia articles pre-training baseline.
5. The results show that the proposed solution performs comparably or better than the current solutions/baselines.
6. Overall the paper is well written and well structured. The authors corroborate all contributions outlined in the introduction section.


**Reasons To Reject:**

1. The authors did not provide any motivation for carrying out token filtering in the experiments section. Was it done due to the observations from Table 4, row/section 4 experiments?
2. In section 4.3, the authors carry out experiments to evaluate if there was any information gain and the results show a significant drop the in the performance of the proposed pipeline. Why was this approach not considered in the first place? The difference clearly shows that using generative LM's for dataset creation can lead to leakage. It would be nice if the authors can discuss additional measures to handle this problem in a closed-domain setting.
3. Why was the number of entities increased from 1 to 10 in entity based dataset generation? What was the motivation for choosing the same?


**Reproducibility:**

3: Could reproduce the results with some difficulty. The settings of parameters are underspecified or subjectively determined; the training/evaluation data are not widely available.

**Reviewer Confidence:**

3: Pretty sure, but there's a chance I missed something. Although I have a good feel for this area in general, I did not carefully check the paper's details, e.g., the math, experimental design, or novelty.

---

> ### Author Rebuttal · Authors · 2023-08-29
>
> We thank the reviewer, for the time they took to write a detailed critique of our work and provide us with encouraging feedback!
>
> 1. The authors did not provide any motivation for carrying out token filtering in the experiments section. Was it done due to the observations from Table 4, row/section 4 experiments?
>
> - During a manual inspection, we identified a small amount of noisy entities special characters (*, !, etc) and specific patterns (https) among others. Training RoBERTa on this _filtered dataset_ yields the highest F1 score on COVID-QA in our study (Table 4), but the impact is not universal. We conclude that entity filtering is likely to have a positive impact on downstream performance if done correctly, but perhaps our filter need be improved. We will include this discussion a bit more in the camera-ready version.
>
> 2. In section 4.3, the authors carry out experiments to evaluate if there was any information gain and the results show a significant drop the in the performance of the proposed pipeline. Why was this approach not considered in the first place? The difference clearly shows that using generative LM's for dataset creation can lead to leakage. It would be nice if the authors can discuss additional measures to handle this problem in a closed-domain setting.
>
> - Although the results in Section 4.3 (ablation study) are lower than the highest scores in Table 4 (main results), we believe this is primarily due to the train/test split generated for the ablation study, rather than information leakage. Our modified RoBERTa model experienced a drop of 12.96% and 7.27% in Exact Match (EM) and F1 scores, respectively, when applied to the ablation set with no entity overlap. Interestingly, other models explored also showed a similar decline in performance, with an average decrease of 12.48% and 8.37% in EM and F1 scores, respectively. Although our model's performance declined in the ablation study, this decrease was consistent with many domain-specific models where information leakage is not possible. Therefore, we do not attribute the performance decline to information leakage but rather to the content of the randomly constructed train/test split for the ablation study. Additionally, any potential information leakage would pertain to the entities present in the test set, not the questions asked about those entities. **_In other words, while our model might have been "tipped off" about the entities in the test set, it would not have known the questions asked about those entities_**. In a real-world application, it would not make sense to intentionally limit the amount or type of entities the model can learn about if they are domain-relevant. For example, when training a patient-QA system for a dental office, practitioners can deduce the relevant entities in the dental domain, but not necessarily the questions patients will ask about those entities. Such training with entities that can increase the “knowledge” of these models is actually highly desirable.
>
> 3. Why was the number of entities increased from 1 to 10 in entity based dataset generation? What was the motivation for choosing the same?
>
> - As described in section 3.2.1, we explored the impact that generating ten contexts per entity, instead of just one, has on downstream model performance. The number of entities, however, remained constant. Having the model generate 10 contexts per entity gives the opportunity for more variation in the conveyed information, which may or may not be beneficial for learning the downstream QA task. Results in Table 4 show that providing additional contexts per entity improves performance in some situations, but incurs a significant computational overhead, making it undesirable in most applications.

---

### Official Review · Reviewer_KEWd · 2023-08-06

**Typos Grammar Style And Presentation Improvements:** 1. Line 089, grammar issue.
**Soundness:** 3

**Excitement:**

2: Mediocre: This paper makes marginal contributions (vs non-contemporaneous work), so I would rather not see it in the conference.

**Paper Topic And Main Contributions:**

This paper proposes to re-pretrain PLMs for downstream close-domain EQA tasks with synthetic data generated by prompting generative LLMs. The paper investigates the idea that by prompting generative LLMs with well-designed prompts and named entities from the target domain, the synthetic data would be close to the target domain in terms of content, style and structure. The paper evaluates its proposed method on two benmark datasets and analyzed their performance and identify critical factors.

**Reasons To Accept:**

1. The generative LLMs do not perform well on many close-domain EQA tasks, especially when the domains are critically different. This paper proposes an interesting method to utilize LLMs' generations for re-pretrain the PLMs for downstream tasks by proper prompting.
2. The paper conducts comprehensive experiments and analysis over two datasets and a series models, which is inspiring and valuable for future research.
3. This paper presents several actionable findings for applying similar pipeline for this task.

**Reasons To Reject:**

1. Wiki papges are used as a baseline method to compare the generations' performance. However, the whole wiki articles are noisy and contain contents that are not relevant to the target domain, which has been well known and thus makes the comparison not fair enough. There should be baselines about retrieval methods, and some other domain adaptation and data augmentation methods like back-translation etc.
2. The research presentation, especially the table position and its analysis locate in several different papges, which makes it hard to follow.
3. The quality evaluation and filtering process is not well presented. There should be more details or source code for this essential process.

**Reproducibility:**

3: Could reproduce the results with some difficulty. The settings of parameters are underspecified or subjectively determined; the training/evaluation data are not widely available.

**Reviewer Confidence:**

3: Pretty sure, but there's a chance I missed something. Although I have a good feel for this area in general, I did not carefully check the paper's details, e.g., the math, experimental design, or novelty.

---

> ### Author Rebuttal · Authors · 2023-08-29
>
> The kind words from reviewer 3 are much appreciated and motivate us to do further work!
>
> 1. Wiki pages are used as a baseline method to compare the generations' performance. However, the whole wiki articles are noisy and contain contents that are not relevant to the target domain, which has been well known and thus makes the comparison not fair enough. There should be baselines about retrieval methods, and some other domain adaptation and data augmentation methods like back-translation etc.
>
> - While it is true that Wiki pages can be noisy and may contain some irrelevant content, using them as a baseline for comparison is still valid and important for several reasons. Despite the noise and irrelevant content, Wikipedia is one of the most comprehensive and commonly used text resources globally, which makes the comparison useful (to see how much we are better against that baseline) and broadly applicable. As reported in [1], mBERTu Wiki, which was pre-trained on Maltese Wikipedia data, surpassed the performance of mBERT, thereby proving to be a competitive baseline. This reinforces the argument that Wikipedia is a valuable resource and serves as a common baseline for different problems. **It is important to note that in our study, a Wikipedia baseline was established alongside domain-specific pre-trained models** to assess the influence of content and text structure during domain adaptation. This indicates that while Wikipedia was used as a baseline, it was not the sole benchmark for evaluation. The use of domain-specific pre-trained models in addition to the Wikipedia baseline helps to provide a more comprehensive evaluation of the model's performance. This dual approach enables researchers to assess not only the model's performance against a common baseline but also its ability to adapt to specific domains, thereby providing a well-rounded evaluation of the model's capabilities.
> - The role of back-translation in this context is not entirely clear to us. If the model is trained to accept a question as input and generates a corresponding answer span, it is essentially the same as training an autoregressive language model (AR LLM) for this task. However, as indicated in Table 3, this approach resulted in suboptimal performance. Therefore, the application of back-translation in this scenario may not yield the desired improvements and needs to be carefully considered and evaluated. On the other side, Wikipedia is often used [2,3] because it provides a common ground for comparison across different models and studies. Other methods like back-translation and domain adaptation are indeed valuable and are often used in conjunction with Wikipedia data or other datasets to improve model performance. However, these methods serve different purposes and are not directly comparable to using Wikipedia as a baseline.
>
> 2. The research presentation, especially the table position and its analysis locate in several different pages, which makes it hard to follow.
>
> - Yes. We agree with this. Apologies for the issues which arose due to this. The tables were placed in such a manner to accommodate all of the content in the 8 pages. However, we realize that this makes it difficult to follow the content. In the camera-ready version we will fix this to make for a better presentation where all tables and figures are placed as close to the discussion as feasible.
>
> 3. The quality evaluation and filtering process is not well presented. There should be more details or source code for this essential process.
>
> - We are unsure as to what is meant by “quality” of the generated samples. We have included examples of both positive and negative generations using all prompt styles, as well as the various prompts attempted, in the appendices. We believe this provides a comprehensive overview of the model's performance across different scenarios. We will consider moving some of these in the main text in the camera-ready version.
> - We describe our token filtering process for each dataset in section 3.2.3. We use regular expressions to remove empirically observed noisy entities such as URL’s, DOI’s, keywords like “Authors”, non-ASCII characters, etc. As mentioned in the last line of the section, it was not feasible to conduct an exhaustive experiment with filtering as there were numerous combinations to explore.. Furthermore, since the TF-IDF filtering did not work as well for COVID-QA, we decided to omit it for RadQA. In fact, with just a cursory filtering as done for it, we see that BERT obtains the best results for the non-combined corpora trials in Table 2, row 6. Is the reviewer also suggesting we mention “all” of the regex rules? Including them in the main paper was not feasible due to space constraints. However, if requested, we can certainly add them to the appendix for a more detailed understanding of our filtering process. Finally, as we mention in the abstract, we will release all code and corpora for better clarity and reproducibility in the camera-ready version upon acceptance of the paper.
>
> References
>
> [1] Micallef, Kurt, et al. "Pre-training Data Quality and Quantity for a Low-Resource Language: New Corpus and BERT Models for Maltese." Proceedings of the Third Workshop on Deep Learning for Low-Resource Natural Language Processing. 2022.
>
> [2] Borkakoty, H., & Espinosa-Anke, L. (2023). WIKITIDE: A Wikipedia-Based Timestamped Definition Pairs Dataset. arXiv preprint arXiv:2308.03582.
>
> [3] Jurczyk, T., Deshmane, A., & Choi, J. D. (2018). Analysis of Wikipedia-based corpora for question answering. arXiv preprint arXiv:1801.02073.

---

### Official Review · Reviewer_TY4M · 2023-08-12

**Soundness:** 4

**Excitement:**

4: Strong: This paper deepens the understanding of some phenomenon or lowers the barriers to an existing research direction.

**Paper Topic And Main Contributions:**

This paper proposes a novel method to generate pre-training data for closed domains and demonstrates it's effectiveness by rich experiments, setting up a new SOTA system on the COVID-QA dataset. By using LLM to generate corpora from entities in QA tasks, the system can effectively generate a small but useful corpus from LLMs to pre-train smaller models like BERT. This work provides a novel and effective knowledge distillation method.

**Reasons To Accept:**

The method is novel.
The detailed result shows the effectiveness of their method.

**Reasons To Reject:**

Whether the resulting model will be influenced by LLM's hallucination problem is to be studied.

**Reproducibility:**

4: Could mostly reproduce the results, but there may be some variation because of sample variance or minor variations in their interpretation of the protocol or method.

**Reviewer Confidence:**

4: Quite sure. I tried to check the important points carefully. It's unlikely, though conceivable, that I missed something that should affect my ratings.

---

> ### Author Rebuttal · Authors · 2023-08-29
>
> First of all, we thank the reviewer for appreciating our hard work! The hallucination problem is one that we mention as a drawback with our approach. While examining examples from the generated dataset, we did observe such instances (mentioned in rebuttal 1; point 1). However, as mentioned in the paper, there does not exist, as far as we know, any reliable method to detect such spurious text. We do however plan to use the standard 6.7B version of Galactica in the final version of the paper to see if it reduces hallucination as their authors also claim better performance with it. A 6.7B version was in fact used in one of the cited papers (_Symbolic knowledge distillation_ … [GPT-3 Curie model]) in our work. Ultimately, we view the hallucination problem as orthogonal to our presented work - others [1] are investigating how to make the model generations more faithful, while we try to make the most of what we are given at the time.
>
> References
>
> [1] Ji, Ziwei, et al. "Survey of hallucination in natural language generation." ACM Computing Surveys 55.12 (2023): 1-38.

---

### Official Review · Reviewer_KF6E · 2023-08-14

**Soundness:** 3

**Excitement:**

3: Ambivalent: It has merits (e.g., it reports state-of-the-art results, the idea is nice), but there are key weaknesses (e.g., it describes incremental work), and it can significantly benefit from another round of revision. However, I won't object to accepting it if my co-reviewers champion it.

**Missing References:**

N/A

**Paper Topic And Main Contributions:**

The paper proposes a framework to generate synthetic corpora for pre-training foundational models (FMs). These generated corpora align well with the writing styles and topics of the downstream tasks resulting in better overall performance. The authors have focused the efforts over two extractive question answering datasets of COVID-QA and RadQA.

**Questions For The Authors:**

Please refer to the weaknesses shared above.

**Reasons To Accept:**

The reasons for accepting this paper are outlined below:

1. The paper is well-structured and easily comprehensible.
2. The paper introduces a framework addressing a crucial NLP issue that resonates with researchers in specialized domains like clinical, finance, or legal.
3. The proposed framework is robust, supported by demonstrable results illustrating its effectiveness across various clinical QA downstream tasks. I have some questions regarding the experiment design and results which I've mentioned in the weaknesses below.

-- Updates --
After reading through the rebuttal, I've updated the reproducibility scores but I believe that there are a lot of additions (discussions, analysis) that would be needed to the camera-ready version. Hence, I'm keeping my original score on soundness and excitement metrics.

**Reasons To Reject:**

Some weaknesses of the paper are as follows:
1. The authors haven't shown any qualitative analysis by showing examples to support the hypothesis in the paragraphs starting lines: 449 and 460. Both these hypothesis can be validated by examining the examples and presenting them in the discussion section.
2. The authors haven't shown the results over multiple runs with the mean and standard deviation of the performance over multiple runs.
3. The results observed in line 491 could also be because of bad train : dev : test splits. This would have been mitigated to certain extent if experimentation was performed over multiple runs.
4. The results across the two datasets are not consistent using different strategies. So, if a researcher is applying this framework to a completely new problem - what would be the suggested strategy to start with? Hence, a small recommendation sub-section would also be need in Section 4.
5. Base BERT and RoBERTa models are used for experimentation. Why not further fine-tune their clinically fine-tuned versions for experimentation? Also, any specific reason why longformers / BigBird weren't used for experimentation? These models have consistently shown better performance in clinical domain where global context changes the overall outcome of a question significantly.

**Reproducibility:**

4: Could mostly reproduce the results, but there may be some variation because of sample variance or minor variations in their interpretation of the protocol or method.

**Reviewer Confidence:**

4: Quite sure. I tried to check the important points carefully. It's unlikely, though conceivable, that I missed something that should affect my ratings.

**Typos Grammar Style And Presentation Improvements:**

Actually, I would like to commend the authors for their pristine figures and tables.

---

> ### Author Rebuttal · Authors · 2023-08-29
>
> First of all, we thank reviewer 1 for their helpful comments! Please find our rebuttals to their questions below.
>
> 1. The authors haven't shown any qualitative analysis by showing examples to support the hypothesis in the paragraphs starting lines: 449 and 460. Both these hypothesis can be validated by examining the examples and presenting them in the discussion section.
> - This is true (449). After further examination of the generated samples, we found instances confirming our hypothesis viz. incorrect information/plain noise. Consider the following prompt from the 470k dataset: **“Title: inhibitors”** there were 10 generations for this prompt. We found 2 cases (cannot show here due to space) where the output either degenerated completely or discussed a topic completely irrelevant to covid-related literature (i.e. graph drawing algorithms). As such, at scale, more noise is introduced leading to a decline in performance. We will provide few such examples in the revised paper for better clarity.
> - For the hypothesis on 460, we remove entities based on regular expressions and string length-based measures. The idea was to remove “noisy” entities which would not strengthen a model’s understanding of the domain. However, this filtering assay did not yield the hypothesized results, as presented in row 5 of Table 4, suggesting the filter as ill-formed, or not sophisticated enough to catch the truly irrelevant entities.
>
> 2. The authors haven't shown the results over multiple runs with the mean and standard deviation of the performance over multiple runs.
> - True. However, as mentioned in Appendix G, RoBERTa is extremely stable and produces consistent scores across multiple trials. BERT on the other hand, is less stable & produces minor variations. While we do show improvements with both models, we anticipate the reader will be more interested in RoBERTa owing to its superior performance & stability. That notwithstanding, we will report the results of multiple (~10) runs with their variations in the final camera-ready version, but will be unable to submit the scores before the rebuttal deadline due to time/resource limitations.
>
> 3. The results observed in line 491 could also be because of bad train : dev : test splits. This would have been mitigated to certain extent if experimentation was performed over multiple runs.
> - Line 491 refers to the Wikipedia trials for RadQA. For the corpus obtained from Wiki, we do not have a train/dev/test split, as with all other strategies. While we will certainly re-run the model through this approach, we firmly believe we will carefully demonstrate the more viable explanation given on line 355 (Wikipedia not possessing most of the entities in the dataset either due to being ill-formed/too esoteric).
>
> 4. The results across the two datasets are not consistent using different strategies. So, if a researcher is applying this framework to a completely new problem - what would be the suggested strategy to start with? Hence, a small recommendation sub-section would also be need in Section 4.
> - Excellent point! We discuss this on line 195. While RadQA & COVID-QA discuss “medical” concepts, their makeup is completely different viz. Radiology reports v/s research articles. As such, even with a “domain” we need a careful analysis of writing styles. Our suggestion to researchers wishing to employ our method on a different task/EQA dataset is to _thoroughly study the dataset_ to pick up linguistic patterns that might emerge. We had to do this with RadQA since synthesizing prompts for it was not as straightforward as COVID-QA. For example, If someone wants to apply our framework to say a legal QA dataset like CUAD, we (authors) would start off by identifying legal terminologies and linguistic cues in the dataset such as use of “Article XV Subsection …” and create our prompts accordingly. This further enforces our point of _tailoring the corpus to the desired writing styles._
>
> 5. Base BERT and RoBERTa models are used for experimentation. Why not further fine-tune their clinically fine-tuned versions for experimentation? Also, any specific reason why longformers / BigBird weren't used for experimentation? These models have consistently shown better performance in clinical domain where global context changes the overall outcome of a question significantly.
> - Great point! Our motivation in this paper was not to rely on existing domain-specific pre-trained FM’s. This is because we wanted to show that a one-size-fits-all type of approach isn’t necessarily the best when dealing with such data. While Bio/Sci-BERT etc. have shown great performance, we wanted to show that even without such expensive pre-training (utilizing several GB’s of data) we can quickly adapt an open-domain model with tailormade corpus. As such, further fine-tuning those models would have defeated the point even though theoretically training them further _could_ have led to further gains. Building on this point, we also wanted to show that by using such a small corpus, we can achieve gains either equal to/more than the existing clinical models.
> - As for longformer/BigBird, we did consider their use seeing as they can deal well with long contexts. However, after viewing the results from an almost similar architecture i.e. XLNet (~3 EM | ~9 F1) capable of dealing with extremely long texts, we decided against the use of the mentioned models. However, we appreciate the suggestion and will compare against  them and report the results in the final version of the paper. We expect these to perform similarly to XLNet too.

---

### Meta-Review · Area_Chair_Hpj4 · 2023-09-19

**Recommendation:** 2

**Metareview:**

This paper introduces a method to generate pre-training data customized for downstream tasks by prompting LLMs. Experimental results on two QA tasks demonstrate the effectiveness of the proposed method.

While the reviewers appreciated importance of the problem space, novelty of the proposed method, and comprehensive experiments, they raised the concerns about the clarity of experimental design and result analysis.

---

### Decision · Program_Chairs · 2023-10-07

**Decision:**

Reject

**Comment:**

This paper introduces a method to generate pre-training data customized for downstream tasks by prompting LLMs. Experimental results on two QA tasks demonstrate the effectiveness of the proposed method.

While the reviewers appreciated importance of the problem space, novelty of the proposed method, and comprehensive experiments, they raised the concerns about the clarity of experimental design and result analysis.